# MITgcm-RN v1.0: Modeling the Transport and Fate of Radionuclides Released from Nuclear Power Plants Wastewater in the Global Ocean Using MITgcm_c65i with the Radionuclide Module

Mao Mao[12#], Yujuan Wang[3#*], Peipei Wu[4], Shaojian Huang[2], Zhengcheng Song[5], Yanxu Zhang[1*]

[1]Department of Earth and Environmental Sciences, Tulane University, New Orleans, LA, 70118, USA

[2]School of Atmospheric Sciences, Nanjing University, Nanjing, 210023, China

[3]School of Environmental Science and Engineering, Nanjing University of Information Science and Technology, Nanjing, 210044, China

[4]Scripps Institution of Oceanography, University of California, San Diego, La Jolla, CA 92093, USA

[5]State Key Laboratory of Environmental Geochemistry, Institute of Geochemistry, Chinese Academy of Sciences, Guiyang, 550081, China

[*]Corresponding author: yzhang127@tulane.edu (Yanxu Zhang); yj.wang@nuist.edu.cn (Yujuan Wang)

[#]These authors contributed equally to this work.

## Abstract

Nuclear energy plays an important role in global energy supply and mitigates greenhouse gas emissions. Potential environmental and human health risks are associated with the generated radioactive isotopes in the wastewater, especially the accidental release during natural disasters. However, the long-term transport and fate of these radionuclides remain uncertain. Here we employ a state-of-the-art ocean tracer model (MITgcm) to simulate the transport and fate of tritium, carbon-14, and seven other typical radionuclides in the twenty-first-century ocean. We use the discharge of radioactive wastewater from the Fukushima Daiichi Nuclear Power Station, both during the March 2011 earthquake and tsunami and from the subsequent release of stored wastewater, as a case study. The model indicates that the Kuroshio and North Pacific Current will spread the radionuclides over the whole North Pacific basin after three years. The enduring transport of long-term discharge in the Pacific will expand to other ocean basins by 2050. Accumulation of particle-reactive radionuclides in the sediments will mostly be centered in the northwest Pacific till 2050. This study demonstrates the effectiveness of our modeling tool, which can be broadly applied to assess the transport and fate of other types of radionuclides and other nuclear discharges worldwide.

**Introduction**

Nuclear energy plays a pivotal role in the current global energy landscape, offering a substantial source of low-carbon electricity that significantly contributes to the reduction of greenhouse gas emissions. As of 2023, there are 413 operational nuclear reactors worldwide, providing a combined capacity of approximately 371.5 gigawatts electric (GW(e)) (IAEA, 2024a). This extensive network of nuclear power plants has been instrumental in avoiding over 2 billion tonnes of carbon dioxide emissions annually, underscoring its effectiveness in combating climate change (IAEA, 2021). Despite these advantages, nuclear power generation carries inherent risks, primarily associated with the production and potential release of radionuclides—radioactive isotopes that can have detrimental ecological and human health impacts (Sathiya et al. 2024). The release of these radionuclides, particularly during natural disasters, can lead to environmental contamination and pose health risks to affected populations (Hasegawa et al. 2015).

The Fukushima Daiichi Nuclear Power Station (FDNPS) disaster serves as a stark example of these risks, which was triggered by the massive earthquake and subsequent tsunami that struck Japan on March 11, 2011. The 9.0-magnitude earthquake caused an automatic shutdown of the reactors; however, the subsequent tsunami, with waves exceeding 15 meters, inundated the power station, disabling its cooling systems and leading to a catastrophic meltdown of three reactor cores (Buesseler, 2020). In the aftermath of the disaster, large volumes of water have been continuously pumped into the damaged reactors to cool the nuclear fuel and prevent further overheating, resulting in the accumulation of approximately 1.25 million tons of wastewater contaminated with various radionuclides (Normile, 2021).

The wastewater was processed by the ALPS (Advanced Liquid Processing System) to remove various radionuclides before being stored in steel tanks (Buesseler, 2020), which has a series of components and steps (e.g., iron/carbonate coprecipitation, adsorbent adsorption, and reverse osmosis membrane) (Lehto et al., 2019). However, 71% of the currently stored wastewater still contained radionuclide concentrations that exceeded the regulatory standards, and a total of 61,800 tons of wastewater had a radionuclide concentration 100 times higher than the regulatory standards as of September 2021 (Schneider et al., 2022). These high radionuclide concentrations are mainly due to the low treatment efficiency and system failure of ALPS in the early years. Therefore, a repurification of these wastewaters is planned before release to the Pacific Ocean (Normile, 2021a). However, the ALPS is ineffective in removing $^{14}$C (carbon-14) and $^{3}$H (tritium) (Buesseler, 2020). The measured $^{3}$H concentrations, even after a repurification, can be as high as 2.7-8.2×$10^5$ Bq/L (1 Bq = 1 radioactive decay per second) (TEPCO, 2020a).

Japan has initiated a phased release of the wastewater into the Pacific Ocean, with the first discharge commencing in August 2023 (Normile, 2021a). The release plan spans several decades, with wastewater being gradually diluted and discharged over time. The wastewater is released through an undersea tunnel extending about one kilometer offshore from the FDNPS into the Pacific Ocean (Normile, 2023). The initial phase of discharge lasted for about 17 days, during which approximately 7,800 tons of wastewater were released (Mabon, 2024). As of November 2024, nearly 55,000 tons of wastewater have been discharged in seven rounds (https://www.nippon.com/en/japan-data/h02103/). The radioactive isotopes contained in the wastewater, such as $^{3}$H, $^{14}$C, cesium-137 ($^{137}$Cs), strontium-90 ($^{90}$Sr), and iodine-129 ($^{129}$I), could pose potential ecological and human health risks (Buesseler, 2020). Some of these radionuclides can bioaccumulate in marine organisms, enter the food chain, and ultimately impact human health through seafood consumption (Real et al., 2004; Carvalho, Oliveira, and Malta, 2011; Fisher et al., 2013). For example, $^{129}$I could be bioaccumulated more than $10^5$ times in seaweeds from seawater (Fisher et al., 1999; Fowler and Fisher, 2005).

Since the 2011 FDNPS accident, substantial progress has been made in modeling the dispersion of radionuclides in the ocean as reviewed by Periáñez et al. (2019). Several studies have employed a range of numerical tools—from Lagrangian particle tracking to full 3D ocean general circulation models (OGCMs)—to simulate the transport, dilution, and fate of radioactive contaminants, especially $^{137}$Cs, a

dominant and detectable tracer. Early studies (e.g., Tsumune et al., 2012; 2013; Behrens et al., 2012; Buesseler et al., 2012; Kawamura et al., 2014; Rossi et al., 2013) rapidly assessed the initial dispersion of radionuclides into the North Pacific using observed and estimated discharge rates. These models were mostly focused on short-term, regional dispersion over the first few months to years. Lagrangian approaches (Kawamura et al., 2011; Nakano and Povinec, 2012; Rypina et al., 2013) offered insights into the transport pathways based on real-time ocean current data, but typically lacked representations of radionuclide decay or interaction with biogeochemical components. Subsequent models evolved to include more comprehensive processes, such as sediment interactions (e.g., Maderich et al., 2017; Periáñez et al., 2019), scavenging (Kamidaira et al., 2021), and bioaccumulation (e.g., Nakano and Povinec, 2012; Maderich et al., 2014a; Carvalho, 2018). Some studies, like those by Miyazawa et al. (2013) and Rossi et al. (2013), employed data-assimilating ocean models to improve accuracy in predicting radionuclide dispersion and arrival times in distant regions such as the U.S. West Coast. Recent work has also extended to the long-term simulation of $^3$H release using general circulation models. For example, using COCO4.9, Cauquoin et al. (2025) simulated both the 2011 accident-related release and future planned discharges under varying climate scenarios and horizontal resolutions, emphasizing the role of ocean dynamics and model resolution in controlling $^3$H transport.

Model fidelity is influenced by many numerical and environmental factors, such as model resolution, diffusion parameters, temperature, salinity, wind, tides, and suspended matter sizes. Recent models have tested these parameters to refine simulations can affect the modeling of radionuclide transport and transformation of radionuclides, and they have been tested in recently released models (Kamidaira et al., 2021; Tsumune et al., 2024; Li et al., 2015). The adoption of advanced hydrodynamic models like the Finite Volume Coastal Ocean Model (FVCOM) and the Hybrid Coordinate Ocean Model (HYCOM) has improved the simulation of coastal and oceanic dispersion processes, enhancing predictions of radionuclide behavior in complex marine environments (Giwa et al., 2025). However, the majority of existing studies focused primarily on $^{137}$Cs and $^{134}$Cs due to their detectability and availability in monitoring data. In contrast, relatively fewer efforts have been made to simulate longer-lived and less easily detected radionuclides such as $^3$H, $^{14}$C, $^{60}$Co, $^{106}$Ru, $^{125}$Sb, $^{129}$I, or $^{90}$Sr (Periáñez et al., 2013; Maderich et al., 2014b). Additionally, most models also targeted near-term impacts (up to 5–10 years), with limited projections beyond 2050. Comprehensive assessments that encompass multiple radionuclides and extend projections to the year 2100 using coupled physical–biogeochemical ocean models remain limited. The necessity for such long-term, multi-radionuclide models is increasingly recognized, particularly in light of ongoing and future nuclear activities and the potential impacts of climate change on marine ecosystems. To gaps, our study aims to develop a comprehensive transport and biogeochemical model for multiple major radionuclides discharged from wastewater and assess their long-term fate in the global ocean. We use the FDNPS release as a case study because the emissions are relatively well characterized and publicly documented, especially in comparison to many other nuclear facilities. The availability of detailed release data allows us to systematically trace the transport, biogeochemical transformation, and decay of radionuclides from a known point source. We first conduct a simulation experiment based on the 2011 FDNPS accident after the Great East Japan Earthquake, which involved a well-documented spike emission of radionuclides into the environment (the verification case). This event has been extensively analyzed in previous studies using observed surface concentrations of $^{137}$Cs as a key tracer (e.g., Kamidaira, 2021; Kawamura et al., 2014; Povinec et al., 2013; Rypina et al., 2013; Bailly du Bois et al., 2012). The relatively well-constrained nature of the initial release provides an opportunity to assess the model's ability to reproduce short- to medium-term transport patterns and validate its performance against observational datasets. We then apply the model to the subsequent release of stored wastewater by FDNPS (the prediction case). We consider three possible discharge scenarios to bracket the uncertainties associated with the emissions, ranging from a low-end case where all releases meet national standards to a high-end case where untreated wastewater is released, with an intermediate scenario in between. We develop the radionuclide model based on the MITgcm ocean tracer model (version: c65i, available from https://mitgcm.org/download/other_checkpoints/MITgcm_c65i.tar.gz), driven by Earth system model (IGSM) predictions, and simulate the dispersion, decay, and biogeochemical cycling of key radionuclides through 2100. Our approach incorporates physical transport, radioactive decay, and

interactions with biological particles, providing a comprehensive framework to evaluate the potential
long-term environmental fate of radionuclide discharges.

**Methods**

**Emissions.** In this verification case, we implement a short-term spike emission of $^{137}$Cs following the timeline of the Fukushima accident, assuming a release duration from March 26, 2011, to April 30, 2011, consistent with available TEPCO reports and previous literature (Kawamura et al. 2011; 2014; Kamidaira 2021). The total released amount of $^{137}$Cs directly discharged from the plant site into the ocean is estimated at $4 \times 10^{15}$ Bq, following previous reconstruction from in situ measurement (Kawamura et al. 2011). In addition to the direct discharge, a significant amount of radionuclides— approximately $6.1 \times 10^{15}$ Bq of 137Cs—was released into the atmosphere and later deposited into the ocean via wet and dry deposition, according to the Nuclear and Industrial Safety Agency (NISA) (Mclaughlin, Jones, and Maher 2012). However, due to the high uncertainty associated with the atmospheric transport and deposition processes, and considering that this study focuses on the marine transport and dispersion of radionuclides from well-characterized oceanic sources, the atmospheric deposition pathway is not included in the present modeling experiment. Instead, we restrict our validation to the direct discharge component, which provides a more constrained and robust basis for evaluating model performance in simulating short-term tracer transport in the ocean.

For the prediction case, the emission scenarios of $^3$H, $^{14}$C, and the seven primary even radionuclides (i.e., $^{60}$Co, $^{90}$Sr, $^{137}$Cs, $^{106}$Ru, $^{134}$Cs, $^{125}$Sb, $^{129}$I) are constructed based on pre-release estimates of radionuclide concentrations and tank inventories at the FDNPS in our study. At the time of model design, detailed information on the status of the ALPS-treated water stored in over 1000 tanks at the FDNPS was not yet available. Therefore, the prediction case was developed using estimates derived from publicly accessible datasets before the commencement of discharge. Specifically, we adopted the measured radiation concentration of $^3$H and other radionuclides for 29 tank areas (TEPCO, 2021a) reported by TEPCO and the average concentration of $^{14}$C (42.4 Bq L$^{-1}$) retrieved from 80 tanks from another report (TEPCO, 2020b). The storage volume of each tank area was estimated with the number of tanks integrated from the FDNPS Site Layout (TEPCO, 2021b) and Jilin-1 satellite images taken on April 8, 2021(Chang Guang Satellite, 2021). Two categories of tank capacity (1000 and 2400 m$^3$) were assumed according to the TEPCO (2019) tank type descriptions.

**Table 1. Annual inventories of $^3$H, $^{14}$C, and other radionuclides from FDNPS under different simulation scenarios for the Prediction Case and the TEPCO (2023) report (units: Bq per year).**

| | $^3$H | $^{14}$C | $^{60}$Co | $^{90}$Sr | $^{137}$Cs | $^{106}$Ru | $^{134}$Cs | $^{125}$Sb | $^{129}$I | Total |
|---|---|---|---|---|---|---|---|---|---|---|
| Low-end | $2.2\times10^{13}$ | $4.6\times10^{9}$ | $1.9\times10^{8}$ | $4.3\times10^{8}$ | $2.2\times10^{8}$ | $4.0\times10^{8}$ | $5.5\times10^{7}$ | $1.9\times10^{8}$ | $7.7\times10^{8}$ | $2.2\times10^{13}$ |
| Intermediate | $1.2\times10^{14}$ | $7.8\times10^{9}$ | $3.5\times10^{8}$ | $9.5\times10^{11}$ | $1.6\times10^{9}$ | $6.6\times10^{8}$ | $1.6\times10^{8}$ | $1.7\times10^{9}$ | $2.0\times10^{9}$ | $1.2\times10^{14}$ |
| High-end | $3.6\times10^{14}$ | $2.9\times10^{10}$ | $6.7\times10^{9}$ | $5.8\times10^{13}$ | $1.1\times10^{10}$ | $2.1\times10^{10}$ | $9.1\times10^{9}$ | $2.2\times10^{10}$ | $8.0\times10^{9}$ | $4.2\times10^{14}$ |
| TEPCO 2023 report | $2.2\times10^{13}$ | $4.4\times10^{9}$ | $5.8\times10^{7}$ | $3.4\times10^{7}$ | $9.1\times10^{7}$ | $2.4\times10^{7}$ | $5.0\times10^{6}$ | $2.5\times10^{7}$ | $4.0\times10^{8}$ | $2.2\times10^{13}$ |

We consider three scenarios (Table 1): the low-end scenario assumes all the discharge will meet the national standards of Japan after secondary ALPS treatment (TEPCO, 2021b). In the case that radioactive wastewater contains multiple radionuclides, the sum of the ratios of each radionuclide concentration to the regulatory standard (shortened as summed ratios hereafter) should be less than one (IAEA, 2005). The proportions of the primary seven radionuclides in the low-end discharge are calculated based on the tank groups that meet the standard. We assume the remaining wastewater will be re-purified and the concentrations of individual radionuclides to be proportionally decreased to a unit summed ratio (TEPCO, 2021c). The discharge of $^3$H is set to 22 trillion Bq per year, as claimed by the Japanese government. The discharge of $^{14}$C is estimated based on the average concentration measured

by TEPCO (TEPCO, 2020c). This results in a total release of $2.2 \times 10^{13}$ Bq per year.

The intermediate and high-end scenarios assume discharging the currently stocked radionuclides without secondary purification, but suppose different future production. The current amounts of all radionuclides are calculated from the measured radioactive concentration and estimated volumes of each tank area. The intermediate scenario presumes the production rate in the future thirty years equivalent

to the present (i.e., four times the current amounts altogether), while the high-end scenario presumes a maximal production, applying the maximum concentration for each radionuclide among the tank groups for future projection. This leads to an estimation of $1.2 \times 10^{14}$ Bq per year and $4.2 \times 10^{14}$ Bq per year for the intermediate and high-end scenarios, respectively. For all the scenarios, we assume the discharge to be constant from January 2023 to December 2050.

The ALPS-treated water is discharged through an undersea tunnel located approximately 1.5 km offshore (TEPCO, 2023), where enhanced mixing by open-ocean currents can facilitate a more homogeneous distribution within the initial release area. However, as the coarse spatial resolution of our model does not explicitly resolve nearshore processes, the discharge is represented at the surface grid cell corresponding to the location of FDNPS (37°N, 141°E).

To facilitate a direct comparison with the actual discharge plan, the inventories reported in TEPCO's *Radiological Environmental Impact Assessment Report Regarding the Discharge of ALPS Treated Water into the Sea* (TEPCO, 2023) have also been incorporated into Table 1. The tritium discharge in our low-end scenario is comparable to TEPCO (2023), while the $^{14}$C and $^{129}$I releases are of similar magnitudes. For particle-reactive radionuclides such as $^{60}$Co, $^{90}$Sr, $^{137}$Cs, $^{106}$Ru, $^{134}$Cs, and $^{125}$Sb, our estimates are

approximately one order of magnitude higher than those in TEPCO (2023), reflecting a more conservative assumption for potential release fractions.

**Transport model.** The simulations of radionuclides in the global ocean are based on a three-dimensional Euler-based transport model, the MITgcm (Marshall et al., 1997). The model has a resolution of 2° × 2.5° horizontally with 22 vertical levels. The model is run from 2023 to 2100 with

ocean circulation data from the Integrated Global Systems Model (IGSM) (Sokolov et al., 2005) (Fig. S6-S8). The ocean boundary layer physics and the effects of mesoscale eddies are modeled based on Large et al. (Large et al., 1994) and Gent and McWilliams (Gent and Mcwilliams, 1990), respectively. We start the model with zero initial conditions for the radionuclides to trace the fate of those contained in the released wastewater.

The model simulates the physical advection and dispersal, radioactive decay, the partitioning of dissolved radionuclides onto suspended particulate matter, and the sinking of particulate-bound radionuclides to deeper waters. The radioactive decay is calculated as first-order kinetics (Huestis, 2018):

$$N(t) = N_0 \left(\frac{1}{2}\right)^{-\frac{t}{\tau}} \tag{1}$$

Where $N_0$ is the number of nuclei present at time t = 0 and $\tau$ is the half-life specific to this isotope (Table

2). An instantaneous equilibrium is assumed between the dissolved radionuclides ($C^d$) and particulate radionuclides ($C^p$) in the seawater:

$$\frac{C^p}{C^d} = \frac{k_d}{f_{oc}} POC \tag{2}$$

where $k_d$ is the partition coefficient (IAEA, 2004) (Table 2), $f_{oc}$ is the fraction of organic carbon in suspended particulate matter ($f_{oc}$ = 10%) (Strode et al., 2010), and *POC* is the concentration of

particulate organic carbon. The monthly concentrations of POC are archived from an ocean plankton ecology and biogeochemistry model (the Darwin project; http://darwinproject.mit.edu) within the MITgcm. The sinking flux of particulate radionuclides is assumed to be proportional to the sinking flux of POC:

$$F_{C^p} = F_{POC} \cdot \frac{C^p}{POC} \tag{3}$$

where $F_{C^p}$ and $F_{POC}$ are the sinking fluxes of $C^p$ and $POC$, respectively. The $F_{POC}$ data is also from the DARWIN model.

**Table 2. Decay half-life and partition coefficients for different radionuclides.**

| Radionuclide | $^{60}$Co | $^{90}$Sr | $^{137}$Cs | $^{106}$Ru | $^{134}$Cs | $^{125}$Sb | $^{129}$I | $^{3}$H | $^{14}$C |
|---|---|---|---|---|---|---|---|---|---|
| Decay half-life [year] | 5.27 | 28.9 | 30.1 | 1.02 | 2.07 | 2.76 | $1.57\times10^{7}$ | 12.3 | 5700 |
| $K_d$ [L/kg] | $5\times10^{7}$ | 200 | 2000 | 1000 | 2000 | 4000 | 200 | 1 | -- |

**Results and Discussion**

**Verification Case.** To better evaluate the model's transport capability under a well-constrained release scenario, we draw upon the Great East Japan Earthquake and the following FDNPS accident in March 2011 as a complementary case study. Previous models and observational studies have extensively investigated the transport patterns of radionuclides released into the Pacific Ocean during this event (e.g., Tsumune et al. 2012; Aoyama et al, 2013; Chen et al., 2021; Kumamoto et al., 2019; Smith et al. 2014). In the immediate aftermath of the accident, substantial quantities of radionuclides, including $^{137}$Cs, entered the marine environment through two primary pathways: direct discharge into coastal waters adjacent to Fukushima, and atmospheric release followed by deposition via wet and dry processes. Model studies indicated the complex interplay of the Kuroshio and Oyashio currents, along with mesoscale eddies, influenced the initial dispersion (Uchiyama et al. 2013; Prants et al. 2017). They indicated that within the first few months, the radioactive plume spread eastward, predominantly staying north of the Kuroshio Current, forming a sharp boundary limiting southward dispersion. Through oceanic processes, radionuclide-contaminated waters were transported across the Pacific basin. Observational data revealed that by June 2012, approximately 1.3 years post-accident, Fukushima-derived $^{137}$Cs were detected 1,500 km west of British Columbia, Canada (Smith et al. 2014).

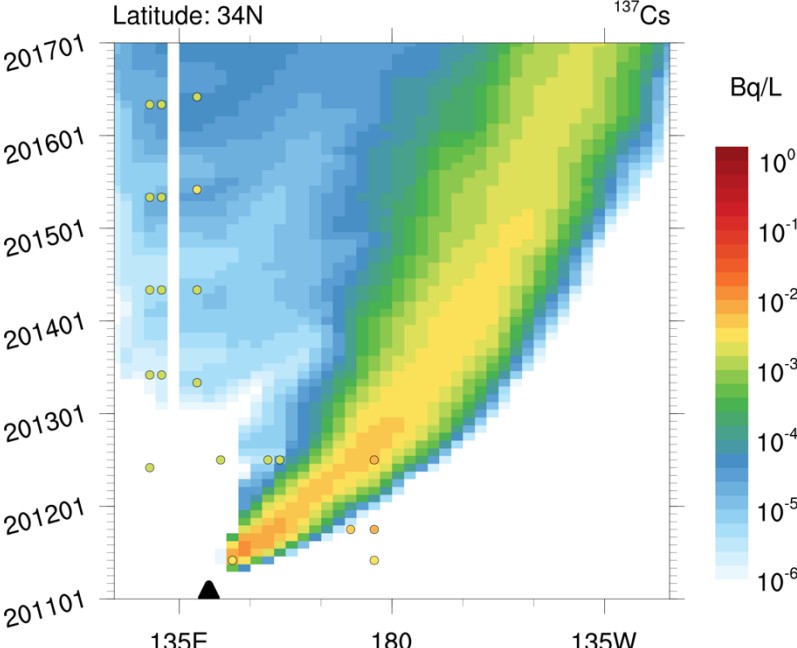

**Fig. 1. Zonal evolution of surface $^{137}$Cs concentration along the latitude of the FDNPS.** The contour plot shows model-simulated concentrations, while overlaid scatter points represent observations from literature (Kaeriyama et al., 2013; Oikawa et al., 2013; Buesseler et al., 2017; Takata et al., 2018; Kenyon et al., 2020). The color scale is set between $10^{-6}$ Bq L$^{-1}$ and 1 Bq L$^{-1}$ to include the observed background levels (~$10^{-3}$ Bq L$^{-1}$) in the North Pacific.

Building upon previous observational and modeling efforts, we quantitatively assess the performance of our model by comparing simulated surface $^{137}$Cs concentrations with available measurements along a latitudinal transect corresponding to the location of the FDNPS in the ocean. The short-term validation period (2011–2016) was chosen to correspond with the time span of most available observational datasets, as extensive monitoring of surface radionuclide concentrations in the North Pacific was conducted during this period. Fig. 1 presents the longitude–time plot of surface $^{137}$Cs concentrations in the North Pacific, illustrating both the observed and simulated spatiotemporal evolution along this transect. The observations indicate that surface $^{137}$Cs concentrations at western Japanese sites such as

Saga and Kagoshima remained at relatively elevated levels (1.7–2.4×10⁻³ Bq L⁻¹) for several years following the initial spike release (Takata et al., 2018). Notably, these values are above the typical background concentration (~1.0×10⁻³ Bq L⁻¹) observed in the North Pacific after 2011. The model gradually approaches these observed concentrations from 2013 onward, suggesting a reasonable representation of long-term tracer transport near the source region.

The simulation results capture the west-to-east advection and subsequent dispersion of ¹³⁷Cs in the months following the accident. Several observations from the central North Pacific in 2011 are included (Kaeriyama et al. 2013). These data show significantly higher ¹³⁷Cs concentrations than simulated, likely reflecting the contribution of atmospheric deposition, which is not considered in the current ocean-only spike release scenario. Another potential reason for the underestimation is the relatively coarse model resolution, which may lead to inaccuracies in reproducing ocean circulation and result in a slower modeled dispersion of the radionuclides, particularly right after the release. While our setup suffers from insufficient regional details, it provides a broader spatial coverage and global perspective essential for evaluating large-scale transport. Previous studies, such as Tsubono et al. (2016), have demonstrated that eddy-resolving regional models incorporating wider atmospheric deposition fluxes could better reproduce radionuclide ¹³⁴Cs dispersion in the Kuroshio region. Nevertheless, observations near the Fukushima coast reported by Oikawa et al. (2013) indicate a surface ¹³⁷Cs concentration of approximately 0.021 Bq L⁻¹ in March and April 2011. The model-simulated concentration at the corresponding grid cell is 0.027 Bq L⁻¹, showing good agreement and building confidence in the accuracy of the initial spike representation in the near-field region.

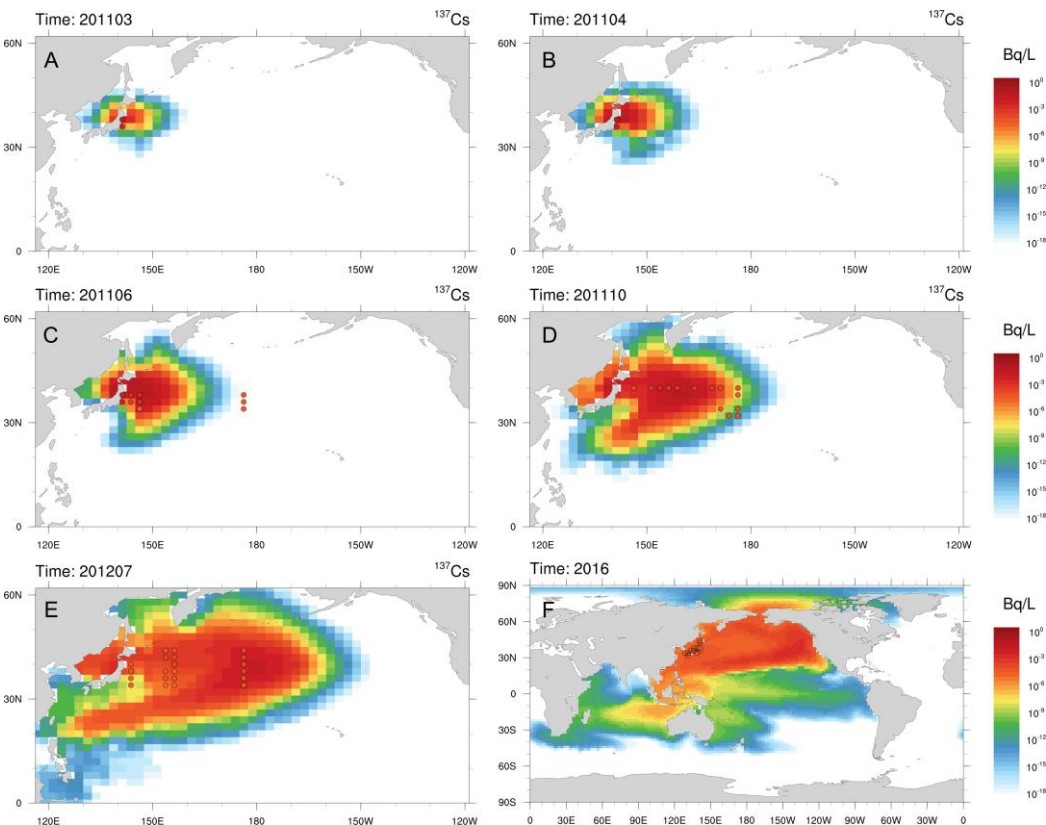

**Fig. 2. Temporal evolution of simulated surface ¹³⁷Cs concentrations from March 2011 to 2016.** Panels A-F show distribution at selected time slices: (A) March 2011, (B) April 2011, (C) June 2011, (D) October 2011, (E) July 2012, and (F) 2016. Panels A–E show the regional North Pacific, while panel F shows the global distribution. The contour plot shows model-simulated concentrations, while overlaid scatter points represent observations from literature (Kaeriyama et al. 2013; Oikawa et al. 2013; Buesseler et al.

2017; Takata et al. 2018; Kenyon et al. 2020).

Our model further reproduces the broader eastward advection pathway of Fukushima-derived radionuclides that has been widely reported in the literature (Fig. 2). Previous model simulations and observational studies (e.g., Smith et al., 2014; Nakano and Povinec, 2012) indicate that the main radioactive plume reached the west coast of North America approximately 2–3 years after the accident, with earlier arrival in northern regions like Alaska and later arrival in southern regions such as California. The simulated eastward progression and eventual entrainment into the North Pacific Gyre facilitate further dispersion toward the subtropical Pacific (Behrens et al., 2012; Rypina et al., 2014), which is consistent with our model results. Fig. 2 shows the simulated evolution of surface $^{137}$Cs concentrations from March 2011 to 2016, together with available observational data. The observed concentrations are generally within the range of $10^{-1}$ to $10^{-3}$ Bq L$^{-1}$, which may in part reflect the detection limits of measurement instruments, in addition to the physical distribution of radionuclides. The observations are mostly located along the Japanese coastal region, indicating the persistence of $^{137}$Cs in nearshore waters, and the model successfully reproduces the regional concentration levels in this area. This agreement supports the model's capability in capturing the long-term transport and retention of radionuclides in the western North Pacific following the initial spike release. However, some discrepancies are found in the central North Pacific. While the model simulates the arrival of the $^{137}$Cs plume at the meridian approximately 6 months after the release—at a concentration level of around $10^{-10}$ Bq L$^{-1}$, Kaeriyama et al. (2013) reported concentrations as high as $5.1 \times 10^{-2}$ Bq L$^{-1}$ at 175°E as early as June 2011. This underestimate may again point to the combined effects of atmospheric deposition, which is not included in the current model setup, and the limited resolution of the model, which may underestimate the speed and extent of transport. The agreement in near-shore regions, along with the identified limitations offshore, helps to characterize the model's capability in capturing the long-term transport and retention of radionuclides in the western North Pacific following the initial spike release.

One interesting evaluation of our model is the transport patterns of the debris entering the Pacific Ocean resulting from the earthquake and subsequent tsunami. This debris weighs approximately 5 million tons, with 70% sinking near Japan's coast and the remainder dispersing across the ocean (Murray et al. 2018). Cumulatively, nearly 100,000 debris items were recorded on North American shores over a four-year period (Murray et al. 2018). Model simulations indicated that this debris began arriving on North American shores approximately two years post-tsunami (Maximenko et al. 2018). The dispersion of this debris was influenced by ocean currents and wind patterns, leading to varied arrival times and locations along North American coastlines. High-windage items, such as buoys and boats, which are more affected by wind due to their protrusion above the water surface, traveled faster and began reaching the Pacific Northwest coast during the winter of 2011-2012, mere months after the tsunami. In contrast, low-windage debris, primarily driven by ocean currents, took a more prolonged period to traverse the Pacific (Murray et al. 2018). Our results for radionuclides in general agree with the low-windage debris, as radionuclides are also transported by ocean currents.

**Prediction Case.** We then apply the model to predict the transport and fate of radionuclides released since 2023 in the global ocean. Similar to the verification case, the influenced area of the proposed release is projected to increase rather fast in the first three years owing to the favorable ocean currents (Fig. 3 and Fig. 4). The modeled total radionuclide concentration in the surface seawater near Fukushima reaches $2.6 \times 10^{-3}$ Bq L$^{-1}$ after the first 90 days of release (high-end scenario for conservativity, the same hereafter except other noted). The energetic current of the Kuroshio, the axis of which is close to Fukushima (141°E, 37°N), carries the released radionuclides eastward by ~2000 km to the middle of the northwest Pacific Ocean. Even though the current speed can be as high as 120-170 cm s$^{-1}$ in this region, the tracer plume spreads much more slowly (~20 cm s$^{-1}$) due to continuous dilution (Lumpkin and Johnson, 2013), which is also influenced by the meandering distribution of eddies and currents (Behrens et al., 2012).

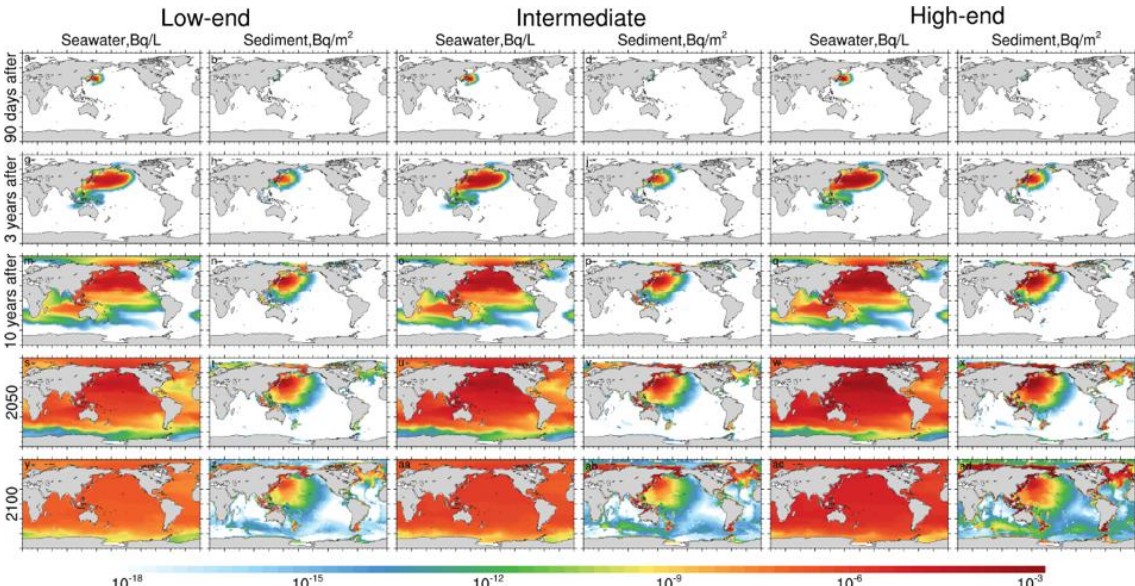

**Fig. 3. Projected total radionuclide concentrations in the seawater (top 10 m) and sediment in the 21st-century ocean.** Data for 90 days, 1 year, 3 years, and 10 years after the initial release date, as well as those in the years 2050 and 2100, are shown. Note that the color scale represents modeled values for comparative purposes, and some values shown in the plots may be below the typical detection limits of observational instruments. The same consideration applies to Figures 4-6.

After one year, the modeled total seawater radionuclide concentration in Fukushima will reach about $6.6$-$7.1\times10^{-3}$ Bq L$^{-1}$ and keep a steady state until 2050 when the proposed release will stop (Fig. 5a). The modeled plume also reaches the dateline in a year and the west coast of North America in 2-3 years carried by the Kuroshio Extension (Sakamoto et al., 2005; Chen et al., 2021). The transport is in general eastward, and the radionuclide concentrations are the highest along the Kuroshio and its Extension (Kumamoto et al., 2019), which forms the west-east axis of the plume. The modeled concentrations at the leading edge of the plume are as high as $1$-$2\times10^{-4}$ Bq L$^{-1}$, a factor of ~10 lower than the source region over Fukushima. A basin-scale influence is simulated after 3 years of continuous release. The North Pacific Current continues to carry the radionuclides to the west coast of North America. The average seawater radionuclide concentrations over the North Pacific Ocean are projected to be $7.4\times10^{-5}$ Bq L$^{-1}$, which is a factor of 100 lower than that in the model grid box near Fukushima and a factor of two lower than the northwest Pacific Ocean ($1.3\times10^{-4}$ Bq L$^{-1}$) (Fig. 5a).

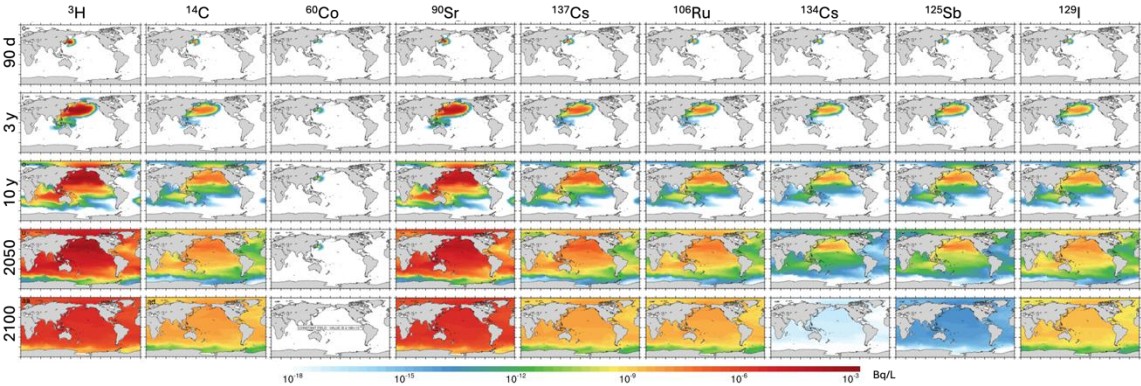

**Fig. 4. Projected spatial distribution of individual radionuclides in the seawater (top 10 m) in the 21st century ocean under the high emission scenario.** Fig. S1 shows the results for the low- and intermediate emission scenarios. Note that the color scale represents modeled values for comparative

purposes, and some values shown in the plots may be below the typical detection limits of observational instruments. For context, background concentrations of radionuclides in the open ocean, based on the MARIS database (https://maris.iaea.org/home), typically fall within the range of $10^{-5}$–$10^{-2}$ Bq L$^{-1}$.

The average total seawater radionuclide concentrations associated with wastewater discharge over North Pacific are projected to increase by a factor of ~5 (to $1.5\times10^{-4}$ Bq L$^{-1}$) and achieve a steady state after 10 years of release (Fig. 5). The concentrations are much higher ($6\text{-}8\times10^{-4}$ Bq L$^{-1}$) in the axis of Kuroshio and over the west coast of North America ($1\text{-}2\times10^{-4}$ Bq L$^{-1}$), where the California and Alaska Currents can expand the plume meridionally. The modeled influence also spreads to other ocean basins, e.g., the Arctic across the Bering Strait and the Indian Ocean by the Indonesian throughflow. However, the modeled radionuclide concentrations over these basins remain low ($2\text{-}6\times10^{-7}$ Bq L$^{-1}$). By 2050 (after 28 years of release), the global average surface ocean radionuclide concentrations are projected to achieve $4.7\times10^{-5}$ Bq L$^{-1}$, while those over the Arctic, Indian, South Pacific, South Atlantic, and North Atlantic Oceans keep increasing even though the concentrations are up to two orders of magnitude lower than the global average. We note that the transport of radionuclides to the Southern Ocean is relatively ineffective due to the blockage of circum-Antarctic currents.

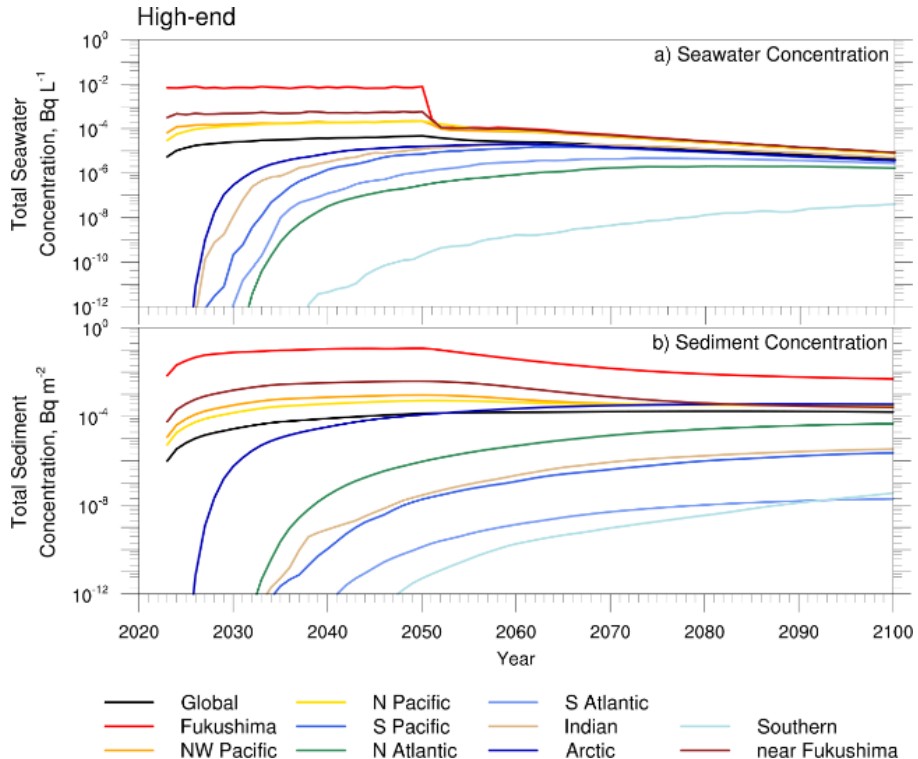

**Fig. 5. Projected total radionuclide concentrations in different ocean basins in the 21st-century ocean**. a) seawater concentrations in the top 10 m and b) vertically integrated concentrations in the sediment. Selected regions and basins include Fukushima (37°N, 141°E), near Fukushima (30-50°N, 132-165°E), Northwest Pacific (EQ-60°N, 130-180°E), North Pacific (EQ-60°N, 130°E -120°W), South Pacific (EQ-60°S, 155°E-80°W), North Atlantic (EQ-60°N, 60-20°W), South Atlantic (EQ-60°S, 40°-10°W), Indian (EQ-60°S, 50-100°E), Arctic (60-90°N), and Southern (60-90°S) Ocean. Note that the y-axis is on a logarithmic scale. Data is for high-end estimates (Fig. S2 shows the results of the low-end and intermediate estimates, respectively). Note that the color scale represents modeled values for comparative purposes, and some values shown in the plots may be below the typical detection limits of observational instruments.

After 2050, at the end of the proposed release, we simulate the seawater concentrations in Fukushima and the nearby regions drop to a level similar to the North Pacific Ocean in 2-5 years (Fig. 3a). The

global average concentration will also begin to decrease due to vertical mixing and the continuous decay of radionuclides. However, the modeled concentrations in other basins will continue to increase due to the continuous transport from the North Pacific Ocean (Chen et al., 2021). The maximum concentrations in these basins are projected to be achieved between 2062 (the South Pacific Ocean) and 2074 (the North Atlantic Ocean), depending on the distance to the source region. One exception is the Southern Ocean, the concentrations of which are modeled to keep increasing by 2100, by when the concentrations in other basins are relatively well mixed within an order of magnitude ($1.7\text{-}7.8\times10^{-6}$ Bq L$^{-1}$).

The primary seven radionuclides account for 15% of the modeled global total concentrations in the first year of release for the high-end release scenario (Fig. S3), with tritium contributing more than 80% (Fig. 2). However, we find that this fraction of the seven radionuclides increases exponentially with time, especially after 2050. In 2100, the fraction is further projected to increase to 52%, with a slightly higher fraction in basins farther away from the North Pacific Ocean basin. This is because many of these radionuclides have a much longer decay half-life ($\tau$) (e.g., 28.1 years for $^{90}$Sr and 16 million years for $^{129}$I) than $^{3}$H (12 years). This highlights the importance of radionuclides other than $^{3}$H in the long term. The transport patterns for the low-end and intermediate-release scenarios are similar but with lower concentration levels, reflecting their lower releases. Given the natural background concentration of tritium ($\sim$50 Bq L$^{-1}$) and the authorized current discharge rates, the simulated signals remain below detectable levels, and no observable anomaly has been identified to date. The modeled concentrations should thus be interpreted as indicative of potential transport pathways rather than as detectable signals.

Some of the primary seven radionuclides (e.g., Co, Cs, and Sr) are particle-reactive and can be adsorbed to sinking particles, which results in an accumulation of the released radionuclides in the sediments, especially over the coastal regions in the northwest Pacific Ocean (Fig. 6). Compared with the seawater, the radionuclides in the sediment are more concentrated in the Northwestern Pacific Ocean close to the source region. The local radionuclide concentrations in Fukushima sediments are modeled to achieve a steady-state of 0.11 Bq m$^{-2}$ in 2050 (Fig. 5b). The modeled average concentrations in the northwest Pacific Ocean sediments are $9.3\times10^{-4}$ Bq m$^{-2}$ in the same year, with much lower concentrations in other basins. Among them, $^{60}$Co has the largest partition coefficient ($K_d = 5\times10^{7}$ L/kg, defined as the ratio of volume concentrations in particulate and dissolved phases), thus dominating the sedimentation flux (50-99% for different scenarios in 2050). However, the relatively short $\tau$ (5 years) of $^{60}$Co limits its transport distance and results in a fast decrease of modeled sedimentation flux away from the northwest Pacific Ocean. The relative contribution to the total sediment concentration is also projected to be exceeded by $^{90}$Sr ($\tau = 28.9$ years) in the mid-2050s. Furthermore, complex redox chemistry could occur to $^{60}$Co that influences its $K_d$ value and biological uptake (Lee and Fisher 1993; Saito and Moffett, 2002). Our results would thus be a simplified estimate of its particulate activity.

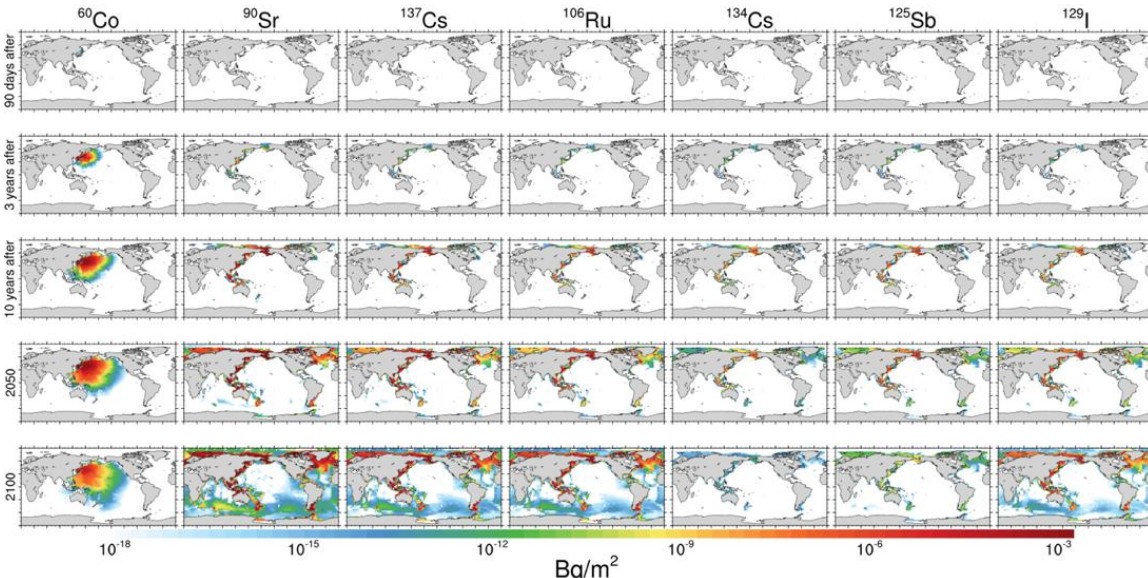

**Fig. 6. Projected spatial distribution of the sediment flux of the primary seven radionuclides in the 21st century ocean under the high emission scenario.** Fig. S4 shows the results for the low- and intermediate emission scenarios. Note that the color scale represents modeled fluxes for comparative purposes. Note that the color scale represents modeled values for comparative purposes, and some values shown in the plots may be below the typical detection limits of observational instruments.

Monitoring programs have been implemented to assess the environmental impact of the released Fukushima wastewater since 2023. Seawater samples are collected from nine designated points near the FDNPS, including locations such as the South and North Water Outlets, and areas 2 km off the Kumagawa and Maeda-Gawa Coasts (https://wwwcms.pref.fukushima.jp). The Japan Nuclear Regulation Authority also has more monitoring sites along the coast, encompassing ~200 km north and south, and expanding ~100 km east into the Pacific Ocean from the FDNPS (https://radioactivity.nra.go.jp). The monitoring focuses on several key radionuclides, including tritium, [137]Cs, and total radionuclides. Sampling is conducted at varying intervals, with rapid analyses performed shortly after discharge events and more detailed assessments carried out monthly. Although the spike release in 2011 during the earthquake and tsunami was high enough to be detected in the downstream regions in the next few years, allowing a direct comparison between observations and model simulations for the verification case, direct comparisons between observational data and model simulations for the prediction case present challenges. Indeed, the estimated [137]Cs emissions during and immediately after the earthquake and tsunami are 5-7 orders of magnitude higher than those from the stored wastewater released since 2023. Observational measurements capture the combined influence of natural background radiation, discharges from various nuclear facilities, and from atmospheric nuclear weapon tests. In contrast, our model simulations focus solely on the radionuclide emissions from the FDNPS, isolating this specific source. There is also no comprehensive background simulation that accounts for both natural sources and long-term anthropogenic emissions of radionuclides in the global ocean.

**Discussions**. Our model simulation is highly demonstrative and subject to large uncertainties. Detailed radionuclide inventories for the ALPS-treated water stored at FDNPS were incomplete at the time of our scenario development. The Japanese government's planning and public reporting have focused primarily on tritium, with less routinely available information for other radionuclides (TEPCO, 2020b). To account for this information gap, we therefore constructed three plausible release scenarios based on the best-available pre-release data and tank-sampling reports. It is noted that TEPCO's *Radiological Environmental Impact Assessment Report Regarding the Discharge of ALPS Treated Water into the Sea* (TEPCO, 2023) provides the officially planned annual discharge inventories, and these values— particularly for non-tritium radionuclides—are generally reported to be below applicable regulatory

limits. While concentrations in the generated raw wastewater before the ALPS treatment contain radionuclide concentrations up to 6 orders of magnitude higher than those after ALPS treatment in the storing tanks (TEPCO, 2020b), the three scenarios in our study span a range of plausible outcomes under current knowledge and are intended to bracket uncertainty for future impact assessment.

The accurate representation of radionuclide distributions near Fukushima is inherently limited by model assumptions and spatial resolution. The assumption of instantaneous mixing within the discharge grid cell and the coarse horizontal resolution ($2° \times 2.5°$), while necessary for global-scale simulations, tends to spread the resulting signals over a large area and smooth out concentration peaks near the discharge pipeline. This limits the model's ability to capture sharp concentration gradients and to compare directly with nearshore observations. According to Japan's dispersion simulations, assuming a discharge concentration of 1500 Bq L$^{-1}$, the modeled $^3$H concentrations rapidly decrease with distance from the outlet, reaching approximately $10^{-2}$ Bq L$^{-1}$ near the outer boundary of their calculation domain ($\sim$490 km $\times$ 270 km) and about $10^{-4}$ Bq L$^{-1}$ at the annual mean level (TEPCO, 2023). These simulated concentrations are 2–4 orders of magnitude lower than the natural tritium concentrations typically observed in the surrounding coastal waters of Japan (approximately 0.1–1 Bq L$^{-1}$). When interpreting the modeled radionuclide concentrations from our work, it is important to consider the background levels of each radionuclide in this region, below which the incremental changes caused by ALPS-treated water discharge would be relatively indistinguishable. These factors highlight the importance of future high-resolution regional modeling for improved near-field validation and more accurate assessment of local-scale dispersion patterns.

Besides the primary seven radionuclides, $^{14}$C, and $^3$H, the stored wastewater contains many other radionuclides (a total of 62, e.g., Yttrium-90, Plutonium-238, Plutonium-239, and Plutonium-240), which contribute 10% to the total wastewater radionuclide concentrations normalized by the individual regulation levels. The measurements for these radionuclides are also less reliable compared with the primary seven, $^{14}$C, and $^3$H, and many of the concentrations are assessed based on proxy data (TEPCO, 2020b). This might cause an underestimation of total excess radioactivity from FDNPS, especially from the strong bioaccumulative ones such as Plutonium-240 (IAEA 2004).

While this study does not provide a quantitative risk assessment, our results establish an essential basis for evaluating potential human exposure. The simulated radionuclide distributions represent the environmental concentrations that determine the external and internal exposure pathways identified in previous studies (Fisher et al., 2013; Jones, 2013). Using a concentration factor (CF) approach (IAEA 2004), the radionuclide concentrations in seafood can be estimated from seawater and sediment concentrations. Human health risk is thus directly related to the radiation doses received, which are measured as the energy absorbed by biological tissues from ionizing radiation (in sievert, Sv). To estimate this dose, the biota radioactivity concentration (e.g., in a unit of Bq L-1) is converted to radiation dose using the dose coefficient (DC), which depends on the type of radioactive decay (e.g., $\alpha$ or $\beta$ type) and the half-life of the radionuclides (Eckerman et al., 2012). The total radiation dose is then calculated by summing the contributions of individual radionuclides, weighted by their respective DC values. Although some radionuclides exhibit relatively high CFs and ingestion DCs, the cumulative effects of long-term exposure to multiple radionuclides remain relatively constrained. Therefore, our model outputs presented here serve as a critical input for future assessments of human and ecological health risks by providing spatiotemporally resolved radionuclide concentrations under different emission scenarios.

We advance beyond previous modeling efforts that primarily utilized inert tracers by explicitly simulating the behavior of specific radionuclides, each characterized by distinct decay rates and biogeochemical interactions. This approach allows for a more accurate representation of radionuclide dynamics in marine environments. Furthermore, our model encompasses multiple environmental media, including seawater and sediment compartments, to capture the complex partitioning and transport processes of radionuclides. By extending our simulations to the end of the 21st century, we provide long-term projections that are crucial for assessing the enduring environmental impacts of radionuclide releases.

As the global community increasingly turns to nuclear energy for its low greenhouse gas emissions, with projections indicating a potential increase in nuclear capacity to 950 GW(e) by 2050 (IAEA, 2024b), ensuring the safety of nuclear power plants becomes paramount to mitigate potential ecological and health risks. A balance between the benefits of nuclear energy in reducing greenhouse gas and other air pollutaemissions with the imperative to minimize ecological and health risks, is required. There is also increasing public apprehension about nuclear plants. Addressing these concerns necessitates comprehensive risk assessments and the development of effective management strategies. While our current study focuses on marine discharges, the modeling framework developed here is broadly applicable to other radionuclide release scenarios with extensive observational data. In particular, the model can be applied to well-documented cases such as global fallout from atmospheric nuclear weapons tests (Nakano and Povinec, 2003; Tsumune et al., 2003; Aoyama and Hirose, 2004) and long-term discharges from European reprocessing plants, notably Sellafield and La Hague Cap (Prandle and Beechey, 1991; Salomon and Breton, 1995). These applications would provide valuable opportunities for further model evaluation and for advancing the understanding of large-scale oceanic dispersion processes. Moreover, similar coupled atmosphere–watershed–ocean modeling approaches could be employed to quantify additional risks associated with atmospheric emissions and deposition of radionuclides—for instance, those originating from cooling processes or nuclear weapons tests (Christoudias and Lelieveld, 2013; Hu et al., 2014; Draxler et al., 2015; Mori et al., 2015). Collectively, these extensions would broaden the applicability of our framework, establishing a foundation for future multi-compartmental radionuclide modeling and comprehensive risk assessments.

**Data availability**

All data generated or analyzed during this study are publicly available at the Zenodo repository: https://doi.org/10.5281/zenodo.16608895 (Mao et al., 2025).

**Code availability**

All model code is archived and accessible via Zenodo at: https://doi.org/10.5281/zenodo.16608895 (Mao et al., 2025). The code is designed to be used in conjunction with the MITgcm (version: c65i, available from https://mitgcm.org/download/other_checkpoints/MITgcm_c65i.tar.gz) and requires proper coupling with its tracer module for execution.

**Author Contributions**

Y.Z. designed this study. M.M., Y.W., P.W., S.H., and Z.S. performed the study. M.M. led the manuscript writing with support from all other authors. M.M. and Y.W., with input from all co-authors, prepared the revision and responses to the reviewers' comments.

**Competing Interests**

The authors declare no competing interests.

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
