# Peer review of "MITgcm-RN v1.0: Modeling the Transport and Fate of Radionuclides Released from Nuclear Power Plants Wastewater in the Global Ocean Using MITgcm\_c65i with the Radionuclide Module"

_EGUsphere, 2025_

## Author Comment (AC1)

**Response to Reviewer 2:**

We would like to thank Reviewer 2 for dedicating time to carefully read our manuscript and provide feedback. We sincerely think their detailed comments have helped us to improve the manuscript. Here it follows a point-by-point response to the reviewer's report (text in black denotes the comments provided, while text in blue denotes our response), associating with the revised manuscript with the track of the changes.

**Comments (in black): General Comments**

This paper employs a state-of-the-art ocean tracer model (MITgcm) to simulate the transport and fate of several radionuclides, including tritium and carbon-14, in the global ocean. The Fukushima Daiichi Nuclear Power Station (FDNPS) accident and the subsequent release of treated water are taken as case studies to evaluate large-scale dispersion by ocean circulation and the accumulation of particle-reactive radionuclides in sediments. The objectives and methodology are appropriate, and I agree with the importance of developing models that account for particle adsorption.

Response (in blue): We appreciate the reviewer's positive assessment of our study and the recognition of the importance of developing models that account for particle adsorption. As the reviewer noted, this work employs a state-of-the-art ocean tracer model (MITgcm) to simulate the global transport and fate of multiple radionuclides, including both conservative (e.g., tritium) and particle-reactive species. By explicitly representing adsorption and settling processes, our model provides an improved framework for assessing radionuclide dispersion and accumulation across different depths. We are pleased that the reviewer found the objectives and methodology appropriate, as our primary aim is to offer a physically consistent and scalable modeling tool for evaluating radionuclide behavior under various release scenarios.

However, there are serious issues with the validation of the Fukushima accident simulation. In addition, regarding the release of ALPS-treated water, since the discharge has already begun, the model should be validated against available observational data before conducting long-term projections. In addition, the current spatial resolution is too coarse, and the assumption of instantaneous dispersion within a release grid cell is inappropriate. In reality, no impact of ALPS-treated water has been detected outside the release cell.

We thank the reviewer for bringing this point up. We totally agree that validation is essential to increase the reliability of model projections. We also realized the conundrum in model validation: the detectable impact of ALPS-treated water is confined to areas near the discharge pipe, yet our model cannot capture such a steep concentration gradient due to coarse resolution. Therefore, we chose to validate the model using the 2011 earthquake and tsunami event, during which the discharges were much higher and produced detectable signals further away from Fukushima, within the range that our model can resolve. In the revision, we added a discussion on

the model's scope and limitations. We clarified the limitations related to model resolution and the assumption of instantaneous mixing within a release cell, and made clear the detectable limit of radioactive signals.

Line 453: "The accurate representation of radionuclide distributions near Fukushima is inherently limited by model assumptions and spatial resolution. The assumption of instantaneous mixing within the discharge grid cell and the coarse horizontal resolution  $(2^{\circ} \times 2.5^{\circ})$ , while necessary for global-scale simulations, tends to spread the resulting signals over a large area and smooth out concentration peaks near the discharge pipeline. This limits the model's ability to capture sharp concentration gradients and to compare directly with nearshore observations. According to Japan's dispersion simulations, assuming a discharge concentration of 1500 Bq L-1, the modeled 3H concentrations rapidly decrease with distance from the outlet, reaching approximately 10 2 Bq L-1 near the outer boundary of their calculation domain ( $\sim$ 490 km  $\times$  270 km) and about 10-4 Bq L-1 at the annual mean level (TEPCO, 2023). These simulated concentrations are 2–4 orders of magnitude lower than the natural tritium concentrations typically observed in the surrounding coastal waters of Japan (approximately 0.1–1 Bq L- 1). When interpreting the modeled radionuclide concentrations from our work, it is important to consider the background levels of each radionuclide in this region, below which the incremental changes caused by ALPS-treated water discharge would be relatively indistinguishable. These factors highlight the importance of future highresolution regional modeling for improved near-field validation and more accurate assessment of local-scale dispersion patterns."

For model application, I recommend using cases where abundant observational data exist, such as atmospheric nuclear weapons tests or discharges from European reprocessing plants. Major revisions are therefore required before this manuscript can be considered for publication.

We appreciate the reviewer's valuable suggestion regarding model applications. As our work focuses primarily on model development and evaluation, the present study establishes a foundation for future applications to various scenarios with abundant observational data. These surely include well-documented cases such as global fallout from atmospheric nuclear weapons tests (Nakano and Povinec, 2003; Tsumune et al., 2003; Aoyama and Hirose, 2004) and long-term discharges from European reprocessing plants, e.g., Sellafield and La Hague Cap (Prandle and Beechey, 1991; Salomon and Breton, 1995). We also rephrased the Discussions section to address the issue.

Line 506: "While our current study focuses on marine discharges, the modeling framework developed here is broadly applicable to other radionuclide release scenarios with extensive observational data. In particular, the model can be applied to well-documented cases such as global fallout from atmospheric nuclear weapons tests (Nakano and Povinec, 2003; Tsumune et al., 2003; Aoyama and Hirose, 2004) and long-term discharges from European reprocessing plants, notably Sellafield and La Hague Cap (Prandle and Beechey, 1991; Salomon and Breton, 1995). These applications would

provide valuable opportunities for further model evaluation and for advancing the understanding of large-scale oceanic dispersion processes. Moreover, similar coupled atmosphere—watershed—ocean modeling approaches could be employed to quantify additional risks associated with atmospheric emissions and deposition of radionuclides—for instance, those originating from cooling processes or nuclear weapons tests (Christoudias and Lelieveld, 2013; Hu et al., 2014; Draxler et al., 2015; Mori et al., 2015). Collectively, these extensions would broaden the applicability of our framework, establishing a foundation for future multi-compartmental radionuclide modeling and comprehensive risk assessments."

**Specific Comments**

**Methods - Emission**

The emission scenario has been summarized in TEPCO's Environmental Impact Assessment Report (TEPCO, 2023), which reflects the actual implementation of ALPS-treated water releases. This report should be cited, and the simulations should be based on actual discharge scenarios rather than pre-release assumptions.

The TEPCO report is available here, starting from page 264: https://www.tepco.co.jp/en/hd/newsroom/press/archives/2023/pdf/230220e0101.pdf

We thank the reviewer for highlighting the importance of using the official discharge plan. At the time of model design, the detailed discharge information was not yet publicly available, and our emission scenarios were therefore constructed based on pre-release estimates derived from tank data and publicly accessible sources. Therefore, **our results serve as a relatively independent estimation of the discharge**. We clarified it in the revised manuscript:

Line 151: "For the prediction case, the emission scenarios of 3H, 14C, and other primary seven radionuclides (i.e., 60Co, 90Sr, 137Cs, 106Ru, 134Cs, 125Sb, 129I) are constructed based on pre-release estimates of radionuclide concentrations and tank inventories at the FDNPS in our study. At the time of model design, detailed information on the status of the ALPS-treated water stored in over 1000 tanks at the FDNPS was not yet available. Therefore, the prediction case was developed using estimates derived from publicly accessible datasets before the commencement of discharge. Specifically, we adopted the measured radiation concentration of 3H and other radionuclides for 29 tank areas (TEPCO, 2021a) reported by TEPCO and the average concentration of 14C (42.4 Bq L-1) retrieved from 80 tanks from another report (TEPCO, 2020b). The storage volume of each tank area was estimated with the number of tanks integrated from the FDNPS Site Layout (TEPCO, 2021b) and Jilin-1 satellite images taken on April 8, 2021(Chang Guang Satellite, 2021). Two categories of tank capacity (1000 and 2400 m3) were assumed according to the TEPCO (2019) tank type descriptions."

We also cited TEPCO (2023) and incorporated the discharge inventories reported therein into Table 1 for direct comparison with our simulated scenarios. The TEPCO (2023) data

represent the officially planned annual discharges, allowing a clearer evaluation of how our pre-release scenarios relate to actual estimates. As shown in Table 1, the tritium discharge in our low-end scenario is comparable to TEPCO (2023), while the 14C and 129I inventories are of similar magnitudes. For other radionuclides (60Co, 90Sr, 137Cs, 106Ru, 134Cs, and 125Sb), our estimates are approximately one order of magnitude higher than those reported by TEPCO (2023).

Line 190: "To facilitate a direct comparison with the actual discharge plan, the inventories reported in TEPCO's *Radiological Environmental Impact Assessment Report Regarding the Discharge of ALPS Treated Water into the Sea* (TEPCO, 2023) have also been incorporated into Table 1. The tritium discharge in our low-end scenario is comparable to TEPCO (2023), while the 14C and 129I releases are of similar magnitudes. For particle-reactive radionuclides such as 60Co, 90Sr, 137Cs, 106Ru, 134Cs, and 125Sb, our estimates are approximately one order of magnitude higher than those in TEPCO (2023), reflecting a more conservative assumption for potential release fractions."

**Results and Discussion**

Line 232: Tsubono et al. (2016) provide a more detailed comparison for this region. Figure 1 is not clear; the authors should refer to Tsubono et al. (2016) and perform a similar validation. Note that Tsubono et al. (2016) used a high-resolution model with relatively good reproducibility of the Kuroshio, whereas the reproducibility in the present model appears poor. The advantages and limitations of the model should be explicitly discussed. Also, Tsubono et al. (2016) applied wider atmospheric deposition fluxes, which should also be referenced.

We appreciate this suggestion. In the revised manuscript, we acknowledged the limitations of our coarser grid in reproducing mesoscale dynamics and not accounting for the wider atmospheric deposition fluxes applied in Tsubono et al. (2016). We also emphasized that our global configuration provides a complementary perspective, offering broader spatial coverage and a consistent framework for assessing long-range radionuclide dispersion:

Line 267: "While our setup sacrifices some regional detail, it provides a broader spatial coverage and global perspective essential for evaluating large-scale transport. Previous studies, such as Tsubono et al. (2016), have demonstrated that eddy-resolving regional models incorporating wider atmospheric deposition fluxes could better reproduce radionuclide 134Cs dispersion in the Kuroshio region."

Figure 1: Since background concentrations of Cs-137 in the North Pacific after 2011 are above 1.0E-3 Bq/L, the color scale used is inappropriate, with the entire comparison range shown as red. The comparison should be made over a range of 1.0E-3 Bq/L to 1 Bq/L.

We agree with this comment. We have revised the color scale of Figure 1 to range from 1.0E-6 Bq/L to 1 Bq/L to more accurately reflect the observer concentration range of Cs-137 in the North Pacific, which is typically between 1.0E-3 Bq/L and 1 Bq/L. This adjustment allows for a more meaningful comparison between simulated and observed concentrations.

Line 242: "Fig. 1. Zonal evolution of surface 137Cs concentration along the latitude of the FDNPS. The contour plot shows model-simulated concentrations, while overlaid scatter points represent observations from literature (Kaeriyama et al., 2013; Oikawa et al., 2013; Buesseler et al., 2017; Takata et al., 2018; Kenyon et al., 2020). The color scale is set between 10-6 Bq L-1 and 1 Bq L-1 to include the observed background levels (~10-3 Bq L-1) in the North Pacific."

We have also updated the manuscript text to clarify the actual range of observed concentrations and to emphasize the background concentration levels.

Line 255: "The observations indicate that surface 137Cs concentrations at western Japanese sites such as Saga and Kagoshima remain at relatively elevated levels (1.7–2.4×10-3 Bq L-1) for several years following the initial spike release (Takata et al., 2018). Notably, these values are above the typical background concentration (~1.0×10-3 Bq L-1) observed in the North Pacific after 2011."

Figure 3: The contour plots are based solely on model output. In reality, the background concentration of tritium is about 50 Bq/L, meaning that the modeled signals are entirely undetectable. The correct conclusion is that the ALPS-treated water signal cannot be detected. Given that dispersion within the release grid cell is a critical limitation, applying this model to ALPS releases is inappropriate.

We thank the reviewer for this important point. We have revised the text to note that, given the natural background of tritium (~50 Bq/L) and the currently discharge rates, the modeled signals remain well below detect limits, consistent with the absence of measurable anomalies to date. Accordingly, we present the contour plots primarily to illustrate possible dispersion pathways and transport patterns under the assumed scenarios, rather than to indicate measurable concentration signals.

Line 396: "Given the natural background concentration of tritium (~50 Bq L-1) and the authorized current discharge rates, the simulated signals remain below detectable levels, and no observable anomaly has been identified to date. The modeled concentrations should thus be interpreted as indicative of potential transport pathways rather than as detectable signals."

We would also like to note that the ALPS-treated water is discharged not at the shoreline but through an undersea tunnel about 1 km offshore (TEPCO, 2023). This offshore release point is subject to stronger mixing and open-ocean currents compared to the immediate coastline, which makes the assumption of relatively uniform mixing within a release grid cell more reasonable under these conditions. This information has now been clarified in the revised manuscript.

Line 185: "The ALPS-treated water is discharged through an undersea tunnel located approximately 1.5 km offshore (TEPCO, 2023), where enhanced mixing by open-ocean currents can facilitate a more homogeneous distribution within the initial release area. However, as the coarse spatial resolution of our model does not explicitly resolve nearshore processes, the discharge is represented at the surface grid cell corresponding to the location of FDNPS (37°N, 141°E)."

At the same time, we acknowledge that our model has limitations in reproducing fine-scale coastal dynamics, and the accuracy near the shoreline is relatively poor. This limitation has been explicitly stated in the manuscript (see Line 446). Our model results are expected to be much more reliable in the open oceans, particularly at a global scale. While the instantaneous mixing of discharge within the coastal model grid cell may have led to an overestimation of the radionuclide transport from the coast to the open ocean, this yields a conservative estimate of global impacts, which is often desirable in environmental pollution modeling.

Figure 4: Other radionuclides should also be compared against the present concentration levels. The MARIS database should be used as a reference: https://maris.iaea.org/home

We appreciate this helpful suggestion. We cited the IAEA's MARIS database as a reference for present-day radionuclide concentrations in the global ocean. The values in Figure 4 represent simulated dispersion patterns under assumed release scenarios and are therefore not intended for direct quantitative comparison with existing background levels. Nevertheless, we have added a note in the text indicating that, according to the MARIS database, most naturally occurring or residual anthropogenic radionuclide concentrations

in the open ocean are typically within  $10^{-5}$ – $10^{-2}$  Bq L-1, which provides useful context for interpreting the modeled magnitude. It is important to recognize that at these low concentrations, many observational instruments might not be able to reliably detect such low levels, especially for radionuclides other than tritium.

Line 354: "For context, background concentrations of radionuclides in the open ocean, based on the MARIS database (https://maris.iaea.org/home), typically fall within the range of 10-5–10-2 Bq L-1."

Figures 5 and 6: As noted above, these figures show concentration levels that are not realistically observable.

We agree that the modeled concentrations in Figures 5 and 6 are not at levels detectable in practice. In the revised manuscript, we clarified that these results are intended to illustrate the potential dispersion pathways rather than to represent realistic observational concentrations. Corresponding notes have been added to the figure captions and discussion for clarity.

Line 334: "Note that the color scale represents modeled fluxes for comparative purposes, and some values shown in the plots may be below the typical detection limits of observational instruments. The same consideration applies to Figures 4-6."

**Discussion**

Line 401: Please refer to TEPCO (2023). The concentrations of all other radionuclides are below discharge limits.

We thank the reviewer for pointing this out. We revised the discussion accordingly to cite TEPCO (2023) and to note that this report provides the officially planned annual discharge inventories, in which the reported concentrations of non-tritium radionuclides are generally below the applicable regulatory limits.

Line 440: "Our model simulation is highly demonstrative and subjected to large uncertainties. Detailed radionuclide inventories for the ALPS-treated water stored at FDNPS were incomplete at the time of our scenario development. The Japanese government's planning and public reporting have focused primarily on tritium, with less routinely available information for other radionuclides (TEPCO, 2020b). To account for this information gap, we therefore constructed three plausible release scenarios based on the best-available pre-release data and tank-sampling reports. It is noted that TEPCO's Radiological Environmental Impact Assessment Report Regarding the Discharge of ALPS Treated Water into the Sea (TEPCO, 2023) provides the officially planned annual discharge inventories, and these values—particularly for non-tritium radionuclides—are generally reported to be below applicable regulatory limits. While concentrations in the generated raw wastewater before the ALPS treatment contain radionuclide concentrations

up to 6 orders of magnitude higher than those after ALPS treatment in the storing tanks (TEPCO, 2020b), the three scenarios in our study span a range of plausible outcomes under current knowledge and are intended to bracket uncertainty for future impact assessment."

Line 413: The uncertainty of concentration measurements is conservatively evaluated using detection limits (TEPCO, 2023). Furthermore, the IAEA MARIS (2021) does not mention underestimation issues. The citation used here is misleading and academically inappropriate.

We appreciate this correction. We removed the inappropriate citation and revised the section to accurately describe the evaluation of concentration uncertainties. We cited TEPCO (2023) and the IAEA MARIS database as the more appropriate references.

Line 458: "According to Japan's dispersion simulations, with a discharge concentration of 1500 Bq L-1, the modeled 3H concentrations rapidly decrease with distance from the outlet, reaching approximately 10-2 Bq L-1 near the outer boundary of their calculation region and 10-4 Bq L-1 at the annual mean level (TEPCO, 2023). They also point out that these modeled values are 2–4 orders of magnitude lower than the natural tritium concentrations observed in the surrounding coastal waters of Japan (about 0.1–1 Bq L-1)."

---

## Author Comment (AC2)

**Response to Reviewer 1:**

We would like to thank Reviewer 1 for dedicating time to carefully read our manuscript and provide feedback. We sincerely think their detailed comments have helped us to improve the manuscript. Here it follows a point-by-point response to the reviewer's report (text in black denotes the comments provided, while text in blue denotes our response), associating with the revised manuscript with the track of the changes.

Comments (in black): The study investigated the transport and long-term fate of radionuclides released from wastewater in the global oceanic environment, using FDNPS release as a case study. The authors applied a more comprehensive transport and biogeochemical model – MITgcm ocean tracer model to run a short- to medium-term predictions for current status and model validation, and then predicted a longer-term fate until 2100 under an intermediate scenario and the low-end and high-end emission scenarios for the uncertainty range. The work is extremely significant, as 1) the health risks posed by the radionuclides released after the earthquake and the ongoing release with intentional wastewater discharge is of concern globally; and 2) although previous studies have modelled some radionuclides regionally after the disaster, we obviously would like to know the risks of all major radionuclides in the long-term future in the global ocean, considering the continuous release. This study used a model capable to consider multiple radionuclides and their essential biogeochemical processes, which is important and interesting.

**Response (in blue):** We sincerely thank the reviewer for the positive and constructive comments. Below we provide point-by-point responses to the comments and indicate the corresponding revisions made in the manuscript.

The manuscript is well organized and prepared. The model performance is validated reasonably using observational data. The authors clearly showed the temporal and spatial pattern of radionuclides. The detailed questions below should be addressed before acceptance.

1. For the emissions in the method section, is there any reference for the assumption that the discharge will last until 2050? Is the assumption more likely to be for a conservative or aggressive assessment?

We appreciate this comment. The assumption of continuous discharge until 2050 is based on the Japanese government's announcement in 2021 that the estimated release of treated wastewater was expected to last for about 30 years (https://www.cnbc.com/2021/04/13/japan-to-release-water-from-fukushima-nuclear-plant-into-sea-in-2-years.html?msockid=2952ddfd6d92623923f5c9b56cb26327). Compared with the actual discharge scenario described in TEPCO (2023) Environmental Impact Assessment Report, as noted by reviewer #2, our setup can be regarded as a more conservative (upper-bound) estimate intended to capture the potential long-term impact.

This approach ensures that the uncertainty range covers the possible maximum duration of releases. We added the Methods section to explicitly clarify the rationale.

Line 190: "To facilitate a direct comparison with the actual discharge plan, the inventories reported in TEPCO's *Radiological Environmental Impact Assessment Report Regarding the Discharge of ALPS Treated Water into the Sea* (TEPCO, 2023) have also been incorporated into Table 1. The tritium discharge in our low-end scenario is comparable to TEPCO (2023), while the 14C and 129I releases are of similar magnitudes. For particle-reactive radionuclides such as 60Co, 90Sr, 137Cs, 106Ru, 134Cs, and 125Sb, our estimates are approximately one order of magnitude higher than those in TEPCO (2023), reflecting a more conservative assumption for potential release fractions."

2. Why did the authors choose to simulate the short-term concentration till 2016 for validation? Is this relevant to the sampling year of the available observation data?

Yes, the short-term validation period was chosen to match the time frame of available observational datasets. Most monitoring programs reported surface and subsurface radionuclide concentrations from 2011 to 2016 in the North Pacific (e.g., Aoyama et al., 2013; Smith et al., 2014; Kaeriyama et al., 2016). After 2016, observations became sparse, which limited the opportunity for systematic comparison. We clarified this in the text.

Line 250: "The short-term validation period (2011–2016) was chosen to correspond with the time span of most available observational datasets, as extensive monitoring of surface radionuclide concentrations in the North Pacific was conducted during this period."

3. It is not to say that you have to assess the consequent risks quantitatively, but is it possible to add a bit discussion on the last section. For example, what's the potential short-term and long-term exposure risks. Is there enough toxicological data for assess this for the combined contamination of such many radionuclides. Can we be sure about the risks based on the predications and available toxicological data of radionuclides both in the short-term and long-term exposure?

We agree with the reviewer that it is valuable to briefly discuss potential exposure risks. In the revised manuscript, we added a paragraph in the Disucussions section.

Line 476: "While this study does not provide a quantitative risk assessment, our results establish an essential basis for evaluating potential human exposure. The simulated radionuclide distributions represent the environmental concentrations that determine the external and internal exposure pathways identified in previous studies (Fisher et al., 2013; Jones, 2013). Using a concentration factor (CF) approach (IAEA 2004), the radionuclide concentrations in seafood can be estimated from seawater and sediment concentrations. Human health risk is thus directly related to the radiation doses received, which are measured as the energy absorbed by biological tissues from ionizing radiation (in sievert, Sv). To estimate this dose, the biota radioactivity concentration (e.g. in a unit of Bq L-1) is converted to radiation dose using th dose coefficient (DC), which depends on the type

of radiation decays (e.g.,  $\alpha$  or  $\beta$  type) and the half-life of the radionuclides (Eckerman et al., 2012). The total radiation dose is then calculated by summing the contributions of individual radionuclides, weighted by their respective DC values. Although some radionuclides exhibit relatively high CFs and ingestion DCs, the cumulative effects of long-term exposure to multiple radionuclides remains poorly constrained. Therefore, our model outputs presented here serve as a critical input for future assessments of human and ecological health risks by providing spatiotemporally resolved radionuclide concentrations under different emission scenarios."

4. Please check the language again, as there are minor grammar mistakes, for example, lines 94-96

We carefully proofread the manuscript and corrected grammatical and typographical errors, including those pointed out by the reviewer.

Line 97-101: "Model fidelity is influenced by numerous numerical and environmental factors such as model resolution, diffusion parameters, temperature, salinity, wind, tides, and particle size distributions. Recent studies have tested these parameters to refine simulations, demonstrating that they can substantially affect the transport and transformation of radionuclides (Kamidaira et al., 2021; Tsumune et al., 2024; Li et al., 2015)."